# Field Control Effect and Initial Mechanism: A Study of Isobavachalcone against Blister Blight Disease

**DOI:** 10.3390/ijms241210225

**Published:** 2023-06-16

**Authors:** Xiuju Yang, Kunqian Cao, Xiaoli Ren, Guangyun Cao, Weizhi Xun, Jiayong Qin, Xia Zhou, Linhong Jin

**Affiliations:** 1National Key Laboratory of Green Pesticide, Key Laboratory of Green Pesticide and Agricultural Bioengineering, Ministry of Education, Guizhou University, Guiyang 550025, China; 2College of Tea Science, Guizhou University, Guiyang 550025, China

**Keywords:** isobavachalcone, blister blight disease, control efficacy, defensive enzymes, microbial diversity, internal transcribed spacer (ITS)

## Abstract

Blister blight (BB) disease is caused by the obligate biotrophic fungal pathogen *Exobasidium vexans* Massee and seriously affects the yield and quality of *Camellia sinensis*. The use of chemical pesticides on tea leaves substantially increases the toxic risks of tea consumption. Botanic fungicide isobavachalcone (IBC) has the potential to control fungal diseases on many crops but has not been used on tea plants. In this study, the field control effects of IBC were evaluated by comparison and in combination with natural elicitor chitosan oligosaccharides (COSs) and the chemical pesticide pyraclostrobin (Py), and the preliminary action mode of IBC was also investigated. The bioassay results for IBC or its combination with COSs showed a remarkable control effect against BB (61.72% and 70.46%). IBC, like COSs, could improve the disease resistance of tea plants by enhancing the activity of tea-plant-related defense enzymes, including polyphenol oxidase (PPO), catalase (CAT), phenylalanine aminolase (PAL), peroxidase (POD), superoxide dismutase (SOD), *β*-1,3-glucanase (Glu), and chitinase enzymes. The fungal community structure and diversity of the diseased tea leaves were examined using Illumina MiSeq sequencing of the internal transcribed spacer (ITS) region of the ribosomal rDNA genes. It was obvious that IBC could significantly alter the species’ richness and the diversity of the fungal community in affected plant sites. This study broadens the application range of IBC and provides an important strategy for the control of BB disease.

## 1. Introduction

The tea plant (*Camellia sinensis*) is considered one of the most important agroeconomic crops because of its wide use as a beverage [1,2,3]. However, the tea plant is easily affected by various diseases during the growth process, resulting in a serious reduction in tea yield [4,5,6,7]. One of the most important tea plant diseases is known as tea tree blister blight (BB), or tea plant leaf swelling. BB is caused by the obligate biotrophic fungal pathogen *Exobasidium vexans* Massee (*E. vexans*), which seriously affects the tea buds and young leaves and causes poor-quality yields and economic losses [8,9,10,11,12,13]. This pathogen *E. vexans* primarily causes pale-yellow translucent spots within the first three to seven days of infection. Then, seven to nine days after the appearance of yellow spots, the disease spots on the adaxial surface of the leaf become sunken and light yellowish brown in color, and the abaxial surface of the infected leaf forms protruding white circular blisters [13]. At present, the control of this tea BB disease is highly dependent on the frequent applications of broad-spectrum fungicides: for example, cupric sulfate [14], thiadiazole copper [15], and pyraclostrobin (Py) [16]. Traditional pesticides can effectively control BB, but their residues can be harmful to the local ecosystem and to the quality of the tea plant [17,18]. Therefore, effective agents such as chitosan oligosaccharides (COSs) and IBC methods to control this disease are necessary to ensure tea production security. Our previous work proved that 5% COSs could limit pest and disease infections in tea gardens and improve the growth and quality of tea [6,19]. COSs were also reported as being capable of reducing BB incidence and maintaining the induced expressions of different defense-related enzymes and total phenol content compared to the control [20]. Other biocontrol agents, including *Pseudomonas fluorescens* Pf1 [17] and *Ochrobactrum anthropi* BMO-111 [8], were effective against BB in tea.

Isobavachalcone (IBC) is an active ingredient from the seed extract of *Psoralea corylifolia* L., which is a traditional herb with various efficient applications [21]. The seed extract of this plant has been found to possess significant antifungal, antibacterial, antitumor, and antidiabetic activities [22], and was traditionally employed for medical purposes. Studies have found that active extracts containing IBC and other content can effectively inhibit fungi [23]. Recently, IBC has been used in agriculture as a botanical fungicide to control plant diseases such as rice blasts [22], apple rot [24], and cucumber bacterial angular spot disease [25]. Furthermore, it has been widely used in the field control of rice blast disease [26]. IBC has an inhibitory effect on pathogens mainly by disrupting the cell wall, cell membrane, mitochondrial membrane, and nuclear membrane of the pathogens, as well as by interfering with cellular metabolic processes [27,28].

Phyllosphere microbial communities are closely related to plant health as well as disease progression [29]. High-throughput sequencing techniques are extensively applied for the comprehensive analysis of the structure of the microbial communities of the phyllosphere microbiome [30,31,32]. An internal transcribed spacer (ITS) is currently used for amplification to analyze phyllosphere fungal communities. ITS1 is located between 18S and 5.8S rRNA genes, is the most widely sequenced DNA region in the molecular ecology of fungi, and has been recommended as the universal fungal barcode sequence [33].

However, as a new plant-derived fungicide, IBC has not been reported for its application in BB disease control and its inhibitory mechanism. Further, changes in fungal community structure and diversity in tea leaves regulated by IBC have not been studied. Therefore, we attempted to focus on this issue and carried out related research. IBC was subjected to a field study to evaluate its function by comparing it with other agents. Physiological analysis of the tea plant and Illumina MiSeq sequencing on the internal transcribed spacer (ITS) region were carried out to determine the possible function mechanism of IBC.

## 2. Results

### 2.1. The Field Control Effect of Isobavachalcone (IBC) on Blister Blight (BB)

The study was designed to investigate the control effects of different agent treatments. Data acquired from the control for the first test were collected in early May 2022, and the plants were measured and calculated, as shown in Figure 1A. All treatment groups including IBC, chitosan oligosaccharides (COSs), and Py showed a significantly lower percent disease index (PDI) compared to CK (PDI was 10.88). The average control effects for the IBC treatment were 56.45%, 63.73%, 51.72%, 45.85%, and 31.93%, respectively, for the five gradient dilution times of 300×, 600×, 900×, 1200×, and 1600× (ratio of dilution). Among these, the 600× dilution performed the best. Both the higher and lower dilution times spoiled the effect rather than improving its efficacy against BB. Similarly, COSs at 300×, 500×, 600×, and 1200× reduced infection by 52.23%, 57.34%, 48.15%, and 27.97%, respectively, when compared to CK. It was determined that the 500× dilution performed the best. Compared to the chemical agent pyraclostrobin (Py), the natural pesticides IBC and COSs at quite low dosages inhibited the infection, and an optimized concentration was determined. These test results showed that IBC at 600× exhibited a greater control effect (63.73%) against BB than the other treatment groups, and this was not significantly different from the chemical pesticide Py at 1100× (64.88%). Figure 2 illustrates that the plants treated with two 2.25 g/Ha foliar applications of IBC showed significantly lower disease severity compared to the untreated plants.

To confirm the results, we used the best-performing ratios of IBC and COSs (600× and 500×, respectively) in combination and conducted a similar field test in early June 2022 in the same tea garden. This time, we used Py at 1100× as the positive control. The results of the test are shown in Figure 1B. The average control effect was 61.70% for the IBC (600×) treatment compared to CK (PDI was 11.10). The average control effect was 53.13% for COSs (500×) compared to CK. The combination of both natural agents worked best, and the average control effect was 70.46% compared to CK. The average control effect was 62.58% for the dilution times of Py at 1100× compared to CK. These results indicate a consistent effect for all treatments, and the novel combination of IBC with COSs could enhance its performance.

### 2.2. Relevant Enzyme Activity Results

The activities of polyphenol oxidase (PPO), phenylalanine aminolase (PAL), superoxide dismutase (SOD), catalase (CAT), peroxidase (POD), *β*-1,3-glucanase (Glu), and chitinase enzyme in the tea were investigated, as shown in Figure 3. Compared to the blank group (water-treated healthy plant), a significant reduction was observed in the activities of the different defense enzymes—namely, PPO (0.0.58-fold), CAT (0.43-fold), POD (0.82-fold), SOD (0.63-fold), and Glu (0.76-fold) in the CK (water-treated infected plant) group—while the activities of PAL and chitinase enzymes were slightly up-regulated, although with no significance. Compared to the CK group, almost all of the activities of the defense enzymes had increased in the different treatments, and the combination of IBC and COSs performed better than IBC alone. In the IBC treatment, for example, there were significant increases in the different defense enzyme activities: namely, PPO (1.86-fold), POD (1.47-fold), SOD (1.08-fold), Glu (1.46-fold), and chitinase (1.32-fold). The combined use of IBC and COSs significantly increased in all tested defense enzyme activities: namely, PPO (2.25-fold), CAT (1.83-fold), PAL (1.25-fold), POD (1.44-fold), SOD (1.66-fold), Glu (1.58-fold), and chitinase (1.53-fold). The results showed that the BB infection caused a deterioration in the performance of several defense enzymes. The treatment using natural agents could up-regulate the activity levels of the tested defense enzymes in tea leaves, and some performed even better than those of healthy tea plants. The increase in enzyme activity was enhanced or magnified when the combination of IBC and COSs was applied, thus improving resistance in tea plants.

### 2.3. Results and Analysis of the Fungal Community Diversity

#### 2.3.1. Quality of the Fungal Sequence Data

Analysis of the phyllosphere microorganisms in tea plants was carried out using 188 Illumina NovaSeq 6000 via sequencing of the ITS rDNA gene. Following quality control processing and de-noising, a total of 755,353 fungal sequences and 14,598,995 fungal bases were obtained from 12 samples with an average length of 232 nt (Table A1). The sequences were classified into 328 operational taxonomic units (OTUs) at a 97% similarity level. When the number of samples reached more than 12, the species accumulation map showed that it was close to the plateau stage (Figure A1), indicating that the experimental samples were sufficient for estimating the species’ richness and for describing the fungal diversity.

#### 2.3.2. Fungal Operational Taxonomic Unit (OTU) Distribution and Diversity

A total of 328 OTUs were entered into a Venn diagram; 64 OTUs were shared between the four different groups; and CK, IBC, IBC + COSs, and Py groups exclusively had 94, 38, 15, and 18 OTUs, respectively (Figure 4). The number of fungal OTUs in the CK group was 226, which was significantly higher than that for the other treatment groups (112 OTUs for IBC + COSs, 175 OTUs for IBC, and 140 OTUs for Py). Shannon and Simpson both pointed to community diversity, and Chao1 and ACE pointed to community richness. Good’s coverage of all samples was 1.00, indicating that our sequencing results were sufficient to fully estimate the diversity of the fungal community (Table 1). Alpha-diversity analysis showed that the Shannon indices for the CK, IBC, IBC + COSs, and Py groups were 0.95 ± 0.04, 0.54 ± 0.08, 0.32 ± 0.06, and 0.60 ± 0.04, respectively. The Simpson indices for the CK, IBC, IBC + COSs, and Py groups were 0.23 ± 0.01, 0.11 ± 0.02, 0.07 ± 0.01, and 0.15 ± 0.01, respectively. The Chao1 indices for the CK, IBC, IBC + COSs, and Py groups were 152.78 ± 15.48, 143.40 ± 21.53, 85.62 ± 5.09, and 127.01 ± 18.45, respectively. The ACE indices for the CK, IBC, IBC + COSs, and Py groups were 160.12 ± 13.92, 143.66 ± 31.88, 94.59 ± 7.00, and 123.66 ± 2.21, respectively. In summary, the fungal community diversity and richness were reduced in all treatments, and the phyllospheric microorganisms were more sensitive to the combined IBC + COSs treatment.

#### 2.3.3. Fungal Community Composition

The fungal community composition of different treatment groups is shown in Table 2, and the 12 samples belong to four fungal phyla, 21 classes, 54 orders, 110 families, 175 genera, and 213 species. Compared with the CK group, each treatment showed a suppressive effect in the number of fungi at all taxonomic levels. It is noteworthy that the fungal community number of the combined IBC + COSs treatment significantly affected the existence of fungi at all taxonomic levels.

At the phylum level, there were only two dominant phyla, Ascomycota and Basidiomycota, which together form the largest phylum of fungi. In this study, we compared the phyla with the CK group (97.47%), and Ascomycota was dominant in the other three treatments (98.34~98.88%) (Table 3). The relative abundance of Basidiomycota decreased from 0.84% (CK) to 0.67%, 0.17%, and 0.33% for IBC, IBC + COSs, and Py, respectively. Overall, the relative abundance of the dominant Ascomycota was significantly increased by the agent treatment and it maintained its position while the relative abundance of Basidiomycota was diminished by the agents, especially by the IBC + COSs treatment group.

The dominant taxa and their relative abundance in the fungal community for each treatment in phyla and genus level are provided in Table 3. At the phylum level, as in the CK group, the top-10 genera were *Cladosporium* (87.40%), *Colletotrichum* (4.07%), *Unidentified_Pleosporales_*sp. (4.08%), *Epicoccum* (0.26%), *Setophoma* (0.25%), *Tilletiopsis* (0.30%)*, Didymella* (0.26%)*, Apiotrichum* (0.16%)*, Uwebraunia* (0.18%), and *Unidentified* (0.18%). These listed dominant genera belong to phyla Ascomycota, except *Tilletiopsis*, *Apiotrichum,* and *Exobasidium*, which belong to Basidiomycota according to analysis of the 100 most abundant fungus genera in a maximum-likelihood tree (Figure 5). Among them, the predominant *Cladosporium* maintained the highest relative abundance in different treatments ranging from 91.84% to 96.49% and slightly less in CK (87.40%). It can be inferred that at the genus level, the dominant taxa generally sustain the edge. The predominant genus remained at the top, and the others, *Colletotrichum*, *Unidentified_Pleosporales_*sp., *Epicoccum*, *Setophoma,* and *Tilletiopsis*, decreased in the natural agent treatment.

It is worth mentioning that the relative abundance of *Exobasidium* was 0.02, 0.04, and 0.03% in the IBC, IBC + COSs, and Py treatment groups, respectively, and was significantly lower than that in the CK group (0.07%), but there was no significant difference between these treatment groups.

#### 2.3.4. Spatial Distribution of the Fungal Communities

PCoA plots were used to determine the spatial distribution of the fungal communities. Samples from the CK and the chemical pesticide Py treatment groups were separated from the fungal communities of the IBC and IBC + COSs treatment groups, while samples from the IBC and IBC + COSs treatment groups tended to cluster together. The results showed that there were significant differences between the biopesticides and the chemical pesticides in the treatment of fungal communities of diseased leaves (Figure 6).

#### 2.3.5. Significant Differences in the Fungal Communities

To identify the fungal taxa with significant differences in abundance between the samples from different agent treatment groups, we performed biomarker analysis using the LEfSe method. At an LDA threshold of four, a total of 15 fungal clades were statistically significant in the CK, IBC, IBC + COSs, and Py treatment groups. Among the CK, IBC, IBC + COSs, and Py treatment groups, 4, 0, 6, and 5 clades, respectively, showed an abundance advantage (Figure 7). Specifically, Pleosporales (order), Unidentified_Pleosporales sp. (family), *unidentified Pleosporales* sp. (genus), and *Pleosporales*_sp. (species) were enriched in the CK group. Ascomycota (phylum), Dothideomycetes (class), Capnodiales (order), Cladosporiaceae (family), *Cladosporium* (genus), and *Cladosporium*_sp. (species) were enriched in the IBC + COSs treatment group. Sordariomycetes (class), Glomerellales (order), Glomerellaceae (family), *Colletotrichum* (genus), and *Colletotrichum_tropicale* (species) were enriched in the Py treatment group.

## 3. Discussion

In this study, the natural fungicide isobavachalcone (IBC) and its use in combination with chitosan oligosaccharides (COSs) promoted related defense enzyme activity and showed a significant inhibitory effect on blister blight disease in tea plants. In addition, internal transcribed spacer sequencing (ITS) was performed, and the results suggested that IBC and the combined (IBC + COSs) treatment can regulate the phyllosphere microbial community structure by enriching beneficial fungi that might antagonize plant pathogens, which is conducive to improving disease resistance of tea plants. This provides a valuable theoretical basis for the biological control of tea blister blight disease.

As one of the most destructive and widespread tea diseases, blister blight (BB) has a significantly negative impact on the production of tea [5,13]. Studies have shown that *Psoralea corylifolia* L. seed extract has antifungal activities against rice blast fungus; IBC is the major active component and has been widely used in the field [26,28]. This first report on the field application of IBC against tea blister blight proved that foliar sprays of IBC could significantly reduce the incidence and severity of BB. IBC, at low dosages, showed a good control effect (63.73%), which was close to the effect of the chemical pesticide pyraclostrobin (Py) (64.88%). The novel combination of IBC + COSs worked better than IBC alone and reached 70.46%.

Disease-suffering plants will produce various harmful free radicals that disrupt their normal physiological activities. As reactive oxygen species (ROS), toxic free radicals produced by plant cells in response to stress can damage plant cell membranes, oxidize proteins, and cause DNA damage. The defense system of plants also usually produces enzymes to remove the damage from free radicals and can improve plant resistance to disease; for instance, polyphenol oxidase (PPO), phenylalanine aminolase (PAL), superoxide dismutase (SOD), catalase (CAT), and peroxidase (POD), as defense-related enzymes, played an important role in this respect [17,34,35,36]. Chitinase and *β*-1,3-glucanase (Glu) could degrade chitin and *β*-1,3-glucan, which were the main components of the fungal cell walls [37]. The initial mechanism explored the level changes of various defense enzymes. The results indicated that the most relevant defensive enzyme activities were significantly inhibited when suffering from BB disease, and these were then up-regulated after the agent treatment. For example, the high-level activities of PPO, POD, and Glu in healthy plants (blank) were all suppressed in the disease group (CK), while following the application of the agents, they then recovered to a higher level. Among all the treatments, IBC + COSs performed the best in promoting defense enzyme activity. Similarly, *Psoralea corylifolia* L. extract could increase the activity of POD, PPO, PAL, chitinase, and *β*-1,3-glucanase enzymes in cucumber [38] and tobacco [39], indicating that IBC has immuno-inducing activities. Additionally, the wide application of COSs was approved to increase activities of defense enzymes, such as PPO, CAT, PAL, SOD, chitinase, and *β*-1,3-glucanase [17,20]. Thus, their combination (IBC + COSs) is supposed to possess a stronger potential for the regulation of defense enzymes and to improve the resistance of plants to BB.

The phyllosphere microbial community was evaluated using Illumina MiSeq ITS sequencing, and it showed that compared to CK, the number of fungal OTUs in the Shannon, Simpson, Chao1, and ACE indices decreased in the treated groups. The results indicated that the fungi failed to sustain the community richness and diversity in the different treatments. All treatment groups had reduced fungal community diversity and richness, while the mixed IBC + COSs treatment group had a more significant effect on the fungal community diversity and richness. The above results were consistent with the research featuring an ITS analysis that suggested that a biocontrol reduced the phyllosphere fungal diversity in tobacco [40], and another study that reported that both chemical pesticides and biocontrol applications could reduce the tea leaf phyllosphere microbiome [41].

Most plants host diverse communities of microorganisms. The phyla Ascomycota and Basidiomycota are commonly the dominant fungi in phyllosphere microorganisms [42]. The IBC treatment increased the relative abundance of the predominant Ascomycota from 97.47% (CK) to 98.34% and decreased the abundance of Basidiomycota from 0.84% (CK) to 0.67%. The fungi and their abundance at the phyla level in the present research are close to the result in wild tea plant resources [43]. The genera, including *Cladosporium*, *Colletotrichum*, *Unidentified_Pleosporales_*sp., *Tilletiopsis*, *Epicoccum*, and *Setophoma* are dominant in the microbe community of the test tea plant. Most of them concisely decreased after IBC or COSs agent treatment, while the predominant genus *Cladosporium* increased and remained at the top in all treatments. Some plant microbiomes are beneficial to the plant, but others may function as plant pathogens and damage the host plant [42]. The *Cladosporium* genus of Ascomycota exists in common endophytes, plant pathogens, and fungal hyperparasites [44], some of which are proven as potential biocontrol agents for plant diseases [45,46], and some have pathogenic potential [47]. The next dominant genus, *Colletotrichum*, includes hundreds of species generally known as endophytes or plant pathogens, mainly causing foliar diseases on a wide range of important crops [48,49,50,51]. These above genera were also reported as ubiquitous endophytic fungi in tea plants in Changsha Hunan, China [52], and there are some differences with the wild tea plants in Yiyang Hunan, China [43]. Changes in habitat conditions and tea varietal may directly affect microbe composition and the structure of the fungal population [52]. In particular, the *Exobasidium* genus is an obligate biotrophic fungal pathogen of BB disease and the most serious threat to Asian tea cultivation and industry [8], which was significantly suppressed in the treatment.

Our study showed that foliar sprays of IBC and IBC + COSs significantly reduced the disease index of tea BB disease under field conditions. Further exploration of enzyme activity and microbe community diversity carries important implications for how agents seek to relieve the symptom of infected tea leaves. The application of IBC and IBC + COSs can boost the innate ability of tea plants to defend themselves by enhancing the activities of defense response enzymes and regulating phyllosphere microbial communities, thereby reducing disease incidence. However, this up- or down-regulation of fungal communities’ diversity and abundance, or possible synergistic interactions that can also be attributed to the occurrence of complex plant diseases, was not clear and is worth being concerned about in future research. Similar studies indicated that the application of biocontroller, such as *Bacillus megaterium* PB50 [53] and *Bacillus amyloliquefaciens* B1408 [54], can regulate the microbial community composition, thereafter suppressing the pathogen and alleviating symptoms but not providing the possible role of related microbes.

These results showed that IBC or IBC + COSs could control blister blight disease damage, possibly by increasing the activity of tea tree defense enzymes, on the one hand, and by reducing the abundance and diversity of interleaf fungal communities, on the other. This research will improve our understanding of the mode of action of IBC and provide potential management strategies for plant BB diseases caused by *E. vexans.*

## 4. Materials and Methods

### 4.1. Plant Material and Test Agents

The field tests were conducted in a tea garden in Meitan in Guizhou Province, China (106° N, 26.6° E), with an altitude of 1100 m, an average annual temperature of 19.5 °C, and annual relative humidity of 85–95%. Tea plants in these areas suffer from BB every year. The tested tea plants (*Camellia sinensis* var. sinensis, “Fuding Dabai”) were approximately 20 years old and free of pesticide application during the test.

The seed extract of *Psoralea corylifolia* L. (0.2% IBC in microemulsion) was purchased from Shenyang Tongxiang Biopesticide Co., Ltd., Shenyang, China. Isobavachalcone (IBC) and 5% amino-oligosaccharide (COS) aqueous solution were provided by Hainan Zhengye Zhongnong Hi-Tech Co., Ltd., Haikou, China, and 30% pyraclostrobin (Py) suspension was purchased from Zhejiang Wellda Chemical Co., Ltd., Hangzhou, China. Among them, Py, as an effective chemical pesticide, was selected as the positive control, and clear water was selected as the negative control (CK). An electrostatic sprayer (Knapsack Electric Sprayer 3WBD-15; Taizhou Shengshiyuanlin Electromechanical Technology Co., Taizhou, China) with a 15 L tank capacity was used, and three tanks of water (45 kg) were consumed per Mu (1 Mu = 667 m^2^). All agents were applied to the tea plant in accordance with the provisions of the ratio of dilution following the instructions with an expanded scope to establish the most efficient concentration (Table 4). For example, the recommended dilution ratio was around 750–1000 times for IBC and 450–560 times for COSs, where the dilution ratio *n* means one unit volume of solute (the commercial agent to be diluted) with (approximately) *n* unit volumes of solvent. The most efficient concentration for IBC and COSs would also be combined for a further round of field treatments 12 days later to confirm the results.

### 4.2. Field Experiment and Disease Assessment

The garden plot selected for the 1 May 2022 test consisted of 44 sections, in which 11 treatments with four replications were carried out according to the determined field applications of each agent concentration (Table 4). Each section was set in an area of 40 m^2^ (8 m × 5 m), and the tea plant row spacing was approximately 0.8 m.

On 1 May 2022 (early stage of blister blight (BB) incidence), solutions of IBC, COSs, and Py were applied to the tea plants by spraying evenly on both sides of the leaves following the dilution ratio established (Table 4). A second spray treatment was carried out on 6 May 2022. Then, investigations took place on 12 May 2022 (six days after spraying), and the tender tea leaves, including the third and fourth leaves on the tea plants, were collected according to a five-point sampling method. A total of 200 leaves from each test plot were observed and recorded for calculation of the PDI.

A leaf was considered infected if an active blister lesion of any of the developmental stages described below was present. The disease index was scored following the method of Saravanakumar et al. [17] on a six-point scale (where 0 represents no disease, 1 is for 1% of the leaf area affected, 3 for 2–10%, 5 for 11–25%, 7 for 26–50%, and 9 for ≥50% of the leaf area affected); the percent disease index (PDI) was calculated using Formula (1). The disease control effect was calculated using Formula (2) with the acquired PDI.
(1)$$\mathrm{PDI}=\frac{\mathrm{Sum}\;\mathrm{of}\;\mathrm{individual}\;\mathrm{ratings}}{\mathrm{total}\;\mathrm{number}}\times \frac{100}{\mathrm{Maximum}\;\mathrm{disease}}$$
(2)$$\mathrm{Control}\;\mathrm{effect}\;\left(\%\right)=\frac{\mathrm{PDI}\;\mathrm{for}\;\mathrm{control}\;-\;\mathrm{PDI}\;\mathrm{for}\;\mathrm{treatment}}{\mathrm{PDI}\;\mathrm{for}\;\mathrm{control}}\times 100$$

Subsequently, another similar plot experiment (16 sections for four treatments with four replications) was performed during the following month in June 2022 with the established most effective concentration of each agent according to the data obtained in May 2022, in the same tea garden.

### 4.3. Sample Collection

Three days after the second foliar application in June 2022, the diseased leaves (with the same disease period, same leaf position, and free of other diseases and insects) were collected from the treatment plots and CK areas, respectively. In addition, healthy leaves in the CK area were also collected as the blank control. All samples were rapidly frozen in liquid nitrogen and stored in a refrigerator at −80 °C for the defense enzyme test and fungal community composition detection using high-throughput sequencing.

### 4.4. Assay of Defense Enzymes

Activity of the defense enzymes was measured as previously described [17,20]. Leaves with the same severity of the same disease were selected, and the diseased spots were removed. Selected parts of the tea leaves were ground in liquid N_2_, and 0.1 g homogeneous samples were resolved in 1 mL of distilled water in a centrifuge tube and centrifuged at 8000× *g* for 10 min at 4 °C. Three biological replicates were set. The enzyme activity test was based on biological kit methods. The kits were purchased from Solar Biologics in Beijing, China.

### 4.5. DNA Extraction, PCR Amplification, and High-Throughput Sequencing

Samples of diseased tissues for blister blight disease were obtained following the method of Yu et al. [55]. Each of the 0.5 g samples was ground, and the cetyltrimethylammonium bromide (CTAB) (CTAB; Nobleryder, Beijing, China) method was used according to the manufacturer’s protocol to extract the total DNA. The ITS1 region of fungi ITS DNA genes was amplified from the DNA extracts using the fungi-specific primers (1F-F: 5′-CTTGGTCATTTAGAGGAAGTAA-3′ and 1F-R: 5′-GCTGCGTTCTTCATCGATGC-3′). PCR reactions were carried out with 15 µL of Phusion^®^ High-Fidelity PCR Master Mix (New England Biolabs, Ipswich, MA, USA), 2 µM of forward primers, and about 10 ng template DNA. Thermal cycling consisted of initial denaturation at 98 °C for 1 min, followed by 30 cycles of denaturation at 98 °C for 10 s, 50 °C for 30 s, 72 °C for 30 s, and a final extension at 72 °C for 5 min. Following validation through 2% agarose gel electrophoresis, the PCR products were purified using a gel extraction kit (Qiagen, Dusseldorf, Germany) and then pooled in equimolar concentrations. Sequencing libraries were generated using a TruSeq R DNA PCR-Free Sample Preparation kit (Illumina, Santa Clara, CA, USA), following the manufacturer’s recommendations, and qualified using an Agilent 5400 bioanalyzer (Agilent, Santa Clara, CA, USA). The validated libraries were used for sequencing on an Illumina NovaSeq 6000 platform (Illumina, San Diego, CA, USA), according to the standard Illumina procedures by Metware Biotechnology Co., Ltd. (Wuhan, China).

### 4.6. Statistical AnalysisS

The paired-end reads were merged using Flast (V1.2.7) [56]. Sequence analysis was performed using Uparse software (Uparse v7.0.1001, http://drive5.com/uparse/, accessed on 18 August 2013) [57]. Sequences with ≥97% similarity were assigned to the same OTUs. Multiple sequence alignment was conducted using Muscle software (Version 3.8.31) [58]. The Chao1, Shannon, Simpson, and ACE indices in our samples were calculated using Qiime (Version 1.7.0) and displayed using R software (Version 2.15.3). PCoA analysis was carried out using the ade4 package and the ggplot2 package in R software (Version 2.15.3). IBM SPSS Statistics 26 was used to analyze the data. The mean values were compared, and *p*-values at ≤0.05 were considered to be statistically significant. Enzyme activity was plotted using GraphPad Prism 8.0.2 software (* *p* < 0.05, ** *p* < 0.01, and *** *p* < 0.001).

## 5. Conclusions

Field test results showed that IBC treatment markedly suppressed BB in tea caused by *E. vexans* and lowered the PDI from 10.88 (CK) to 3.94, which was reduced by 63.73% when compared to CK, and which was not significantly different from the chemical pesticide Py (64.88%). However, treatment with IBC + COSs protected the tea plants by up to 70.46% during the BB-prevalent period. Results of the physiological study showed that IBC could increase the activities of PPO, POD, SOD, chitinase, and *β*-1,3-glucanase in tea leaves, and IBC + COSs could improve the activity of PPO, CAT, POD, SOD, chitinase, and *β*-1,3-glucanase enzymes. High-throughput sequencing results showed that treatment with IBC or IBC + COSs significantly reduced the diversity and abundance of fungal communities at the diseased site compared to CK. The results initially indicated that IBC or IBC + COSs can better control BB in tea, possibly by increasing the activity of tea tree defense-related enzymes and by regulating the fungal community diversity at the susceptible site. This study offers a preliminary understanding of IBC biocontrol actors and modes of action, and it provides potential sustainable management strategies for tea plant diseases caused by *E. vexans*.

## Figures and Tables

**Figure 1 ijms-24-10225-f001:**
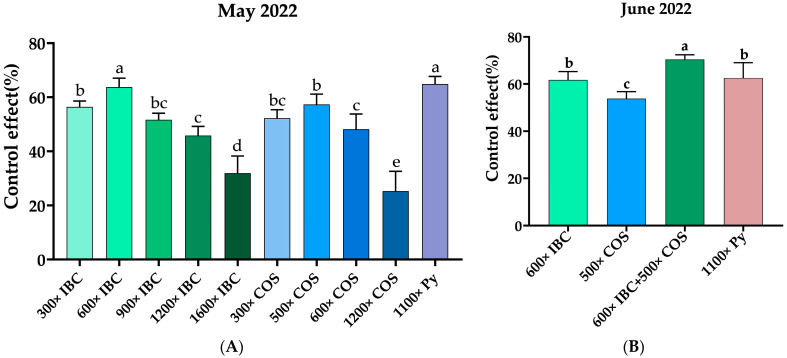
Field control effect of isobavachalcone (IBC) on blister blight disease; (**A**) represents the control effect in May 2022 (PDI for the water control was 10.88); (**B**) represents the control effect in June 2022 (PDI for the water control was 11.10). The data represent the means ± SD of four replicate samples. Different letters indicate significant differences at *p* < 0.05.

**Figure 2 ijms-24-10225-f002:**
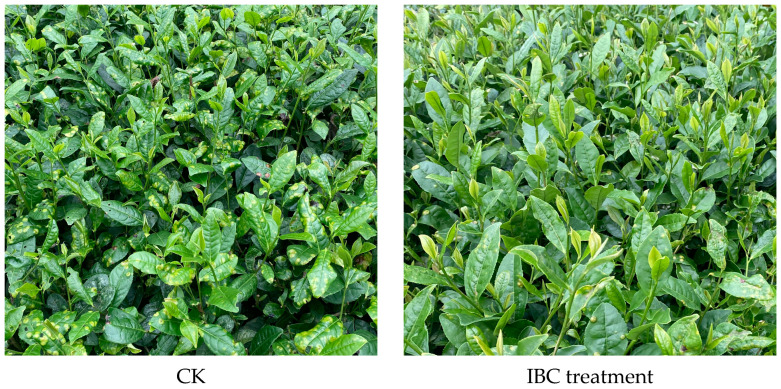
Tea plants treated with IBC showed low levels of infection.

**Figure 3 ijms-24-10225-f003:**
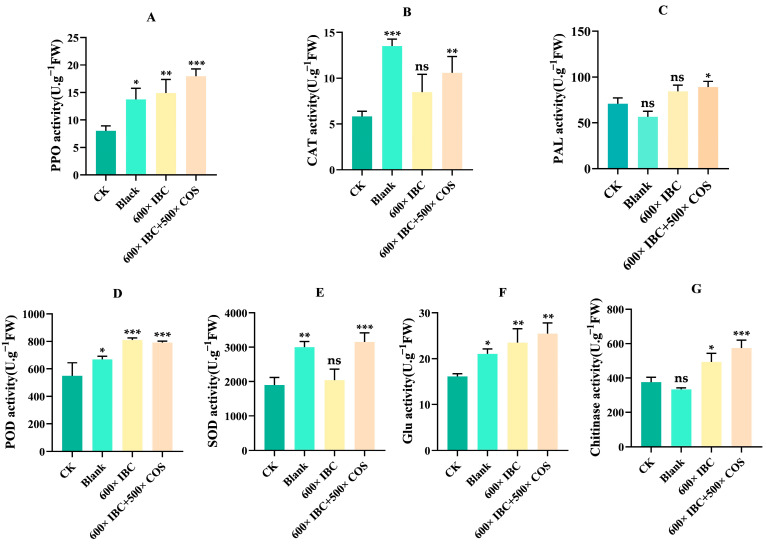
Plots of the relevant defensive enzyme activity results. (**A**) PPO activity; (**B**) CAT activity; (**C**) PAL activity; (**D**) POD activity; (**E**) SOD activity; (**F**) Glu activity; and (**G**) Chitinase activity. CK represents the untreated infection group, and blank represents the untreated healthy group. The data represent the means ± SD of three replicate samples (ns represents no significant difference, * *p* < 0.05, ** *p* < 0.01, and *** *p* < 0.001).

**Figure 4 ijms-24-10225-f004:**
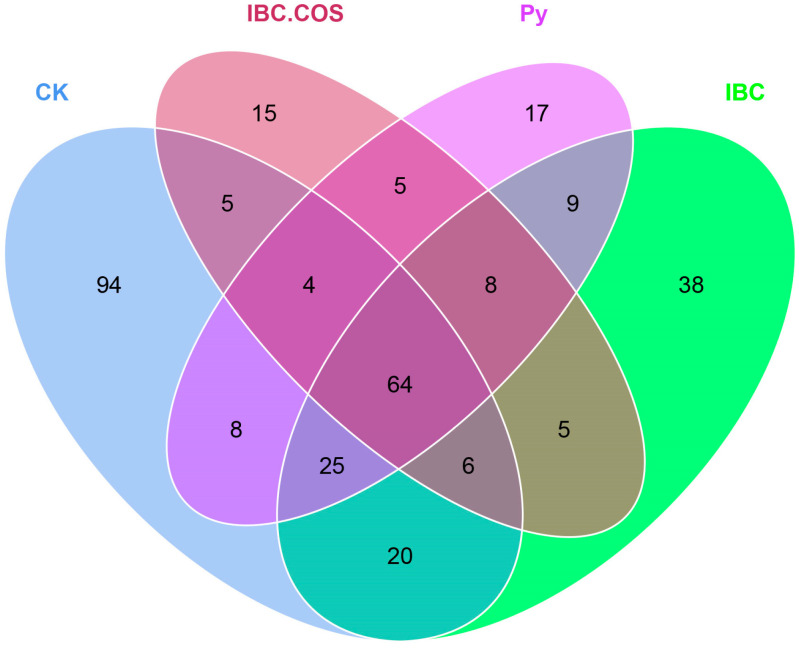
Venn diagram showing the number of fungal OTUs in the different treatments. Numbers in the core section indicate the OTUs shared by each group. Numbers in the overlap areas indicate OTUs unique to two or three samples. Numbers in the non-overlap areas indicate OTUs unique to that group.

**Figure 5 ijms-24-10225-f005:**
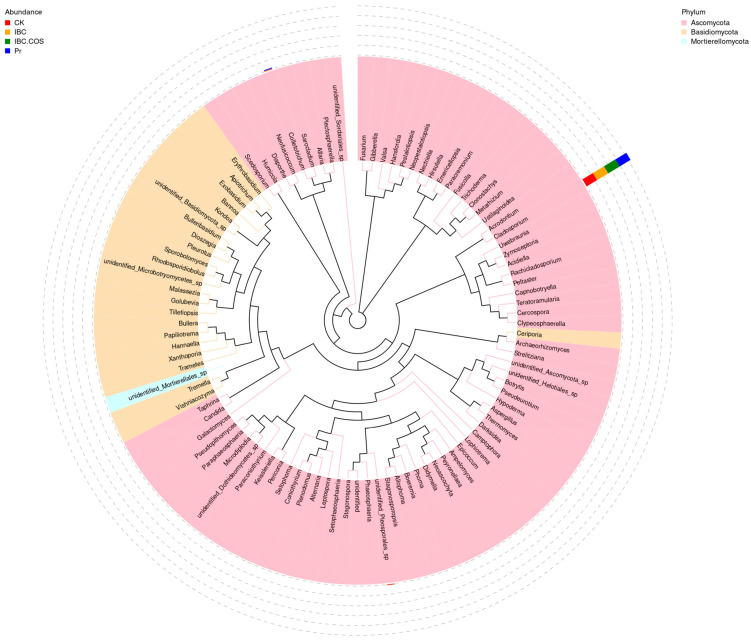
Maximum-likelihood tree of the 100 most abundant genera in the samples from the different treatment groups obtained through the analysis of ITS rDNA data. A color-coded bar plot shows the distribution of each fungal genus in the different groups.

**Figure 6 ijms-24-10225-f006:**
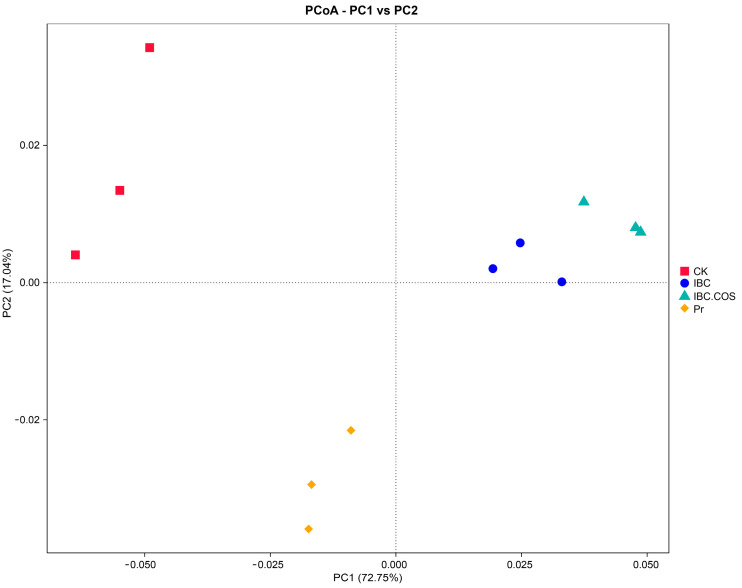
Principal coordinate analysis (PCoA) of the samples with different treatment groups.

**Figure 7 ijms-24-10225-f007:**
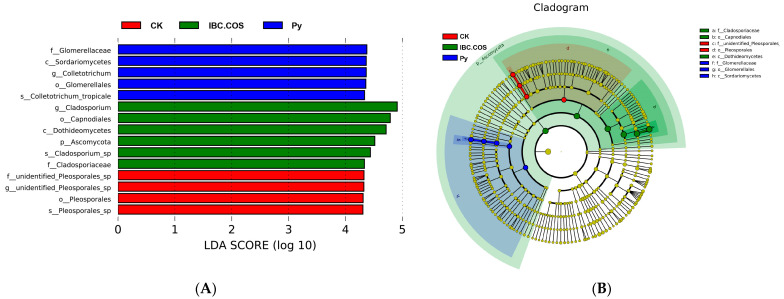
Using the LEfSe method, we analyzed the fungal abundance between the disease samples in the treated and untreated groups; (**A**) shows the LDA score identifying the differentiation sizes between the treatment and control groups of IBC, IBC + COSs, Py, and CK, respectively, with a threshold value of 4; (**B**) shows a cladogram of the fungal communities.

**Table 1 ijms-24-10225-t001:** Alpha-diversity indices of the fungal community based on Illumina MiSeq sequencing of the different samples.

Treatments	Shannon	Simpson	Chao1	ACE	Coverage
CK	0.95 ± 0.04 a	0.23 ± 0.01 a	152.78 ± 15.48 a	160.12 ± 13.92 a	1.00
IBC	0.54 ± 0.08 b	0.11 ± 0.02 c	143.40 ± 21.53 a	143.66 ± 31.88 ab	1.00
IBC + COSs	0.32 ± 0.06 c	0.07 ± 0.01 d	85.62 ± 5.09 b	94.59 ± 7.00 c	1.00
Py	0.60 ± 0.04 b	0.15 ± 0.01 b	127.01 ± 18.45 a	123.66 ± 2.21 bc	1.00

Diversity and richness estimation of the sequencing libraries from the sequencing analysis. The values represent the mean ± SD of three replicates; a *p*-value of <0.05 was statistically significant. The same letters indicate no difference between groups, and different letters (a, b, c, and d) indicate statistically significant differences.

**Table 2 ijms-24-10225-t002:** The number of detected phylotypes classified at different taxonomic levels.

Treatments	Phylum	Class	Order	Family	Genus
CK	3	17	43	86	130
IBC	3	18	38	69	99
IBC + COSs	2	10	27	48	68
Py	2	13	33	62	89
Total No. of detected phylotypes	4	21	54	110	175

**Table 3 ijms-24-10225-t003:** List of the dominant taxa and their relative abundance in the fungal community for each treatment.

Community Structure		Relative Abundance (%)			
		CK	IBC	IBC + COSs	Py
Phylum	Ascomycota	97.47 ± 0.26 c	98.34 ± 0.39 b	98.88 ± 0.18 a	98.55 ± 0.20 ab
	Basidiomycota	0.84 ± 0.18 a	0.67 ± 0.30 a	0.17 ± 0.03 b	0.33 ± 0.02 b
Genus	*Cladosporium*	87.40 ± 0.65 d	94.49 ± 0.86 b	96.49 ± 0.70 a	91.84 ± 0.77 c
	*Colletotrichum*	4.07 ± 0.78 b	1.67 ± 0.35 c	0.87 ± 0.23 c	5.41 ± 0.62 a
	*Unidentified_Pleosporales_*sp.	4.08 ± 0.72 a	0.48 ± 0.07 bc	1.08 ± 0.18 b	0.08 ± 0.02 c
	*Epicoccum*	0.26 ± 0.13 a	0.27 ± 0.13 a	0.05 ± 0.01 b	0.15 ± 0.01 ab
	*Setophoma*	0.25 ± 0.07 a	0.28 ± 0.12 a	0.03 ± 0.01 b	0.17 ± 0.04 a
	*Tilletiopsis*	0.30 ± 0.09 a	0.07 ± 0.04 b	0.03 ± 0.02 b	0.05 ± 0.01 b
	*Didymella*	0.26 ± 0.10 ab	0.31 ± 0.09 a	0.06 ± 0.01 c	0.16 ± 0.04 bc
	*Apiotrichum*	0.16 ± 0.06 ab	0.23 ± 0.11 a	0.02 ± 0.01 c	0.09 ± 0.02 bc
	*Uwebraunia*	0.18 ± 0.04 a	0.07 ± 0.01 b	0.01 ± 0.01 c	0.01 ± 0.00 c
	*Unidentified*	0.18 ± 0.04 a	0.07 ± 0.01 b	0.01 ± 0.01 c	0.01 ± 0.00 c
	*Exobasidium*	0.07 ± 0.04 a	0.02 ± 0.01 b	0.04 ± 0.00 b	0.03 ± 0.01 b

The data in the table are the mean ± SD (*n* = 3). Different lowercase letters in the same column indicate significant differences in treatments at *p* < 0.05.

**Table 4 ijms-24-10225-t004:** Test concentrations for each agent in the diseased plant plot.

Item and Structure of the Active Ingredient	Foliar Application in the Field Treatment		
	Dilution Times (×)	Consumption of Agents per Mu (mL/667 m^2^)	Dosage of Active Ingredient (g/Ha)
Seed extract from *Psoralea corylifolia* L. (0.2% isobavachalcone, IBC) 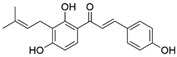	300	150	4.5
	600	75	2.25
	900	50	1.5
	1200	37.5	1.2
	1600	28.12	0.9
5% Chitosan oligosaccharides (COSs) 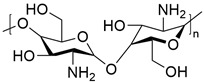	300	150	112.5
	500	90	67.5
	600	75	56.25
	1200	37.5	28.2
30% Pyraclostrobin (Py) 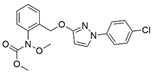	1100	40.9	184.05
Clear water (CK)	/	45 kg	/

## Data Availability

All data generated in this study are presented in the current manuscript. No new datasets were generated. Data are available upon request from the corresponding author.

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
