# Peer review of "Field Control Effect and Initial Mechanism: A Study of Isobavachalcone against Blister Blight Disease"

_ijms, 2023, doi:10.3390/ijms241210225_

Round 1

Reviewer 1 Report (Previous Reviewer 3)

My comments can be found in the attached manuscript. I have to highlight the same suggestions that I made earlier as some of them were not addressed. The results were not discussed well, some parts of the discussion can be found in the introduction and some results are repeated in the discussion.

It still needs some editing and I highlighted a couple of lines.

Author Response

Reviewer 2 Report (Previous Reviewer 1)

Dear colleagues. Please see the attachment.

Author Response

Reviewer 3 Report (Previous Reviewer 2)

The study improved a lot. It can be acceptable.

Author Response

Thanks for your great help in improving our manuscript.

Round 2

Reviewer 1 Report (Previous Reviewer 3)

I have a few more comments, please find them in the attached MS.

There is some improvement compared to previous versions.

Author Response

Reviewer 2 Report (Previous Reviewer 1)

Dear colleagues. Thank you for taking into account the comments

Author Response

Thanks for your great help in improving our manuscript.

This manuscript is a resubmission of an earlier submission. The following is a list of the peer review reports and author responses from that submission.

Round 1

Reviewer 1 Report

Dear colleagues. There are questions and comments about your article.

Reviewer 2 Report

The study is aimed to investigate the control effects of isobavachalcone  in field study, and study the preliminary mechanism of action. The study has some interesting points that can be worth to publish. However, there are several parts of the study that need improvements before possible consideration for publication.

Suggestions:

L18-19: In this study, the control effects of IBC were evaluated on field, and the preliminary mechanism of action was also investigated.

L23: delete 'Surprisingly'

The Introduction is too short. Give more backgound of the lacking information on the study aims.

L47: Give a more precise objective.

L59: Figure 1 title: Give in full IBC, BB.

L80: Text in the figures are too small.

Figure 4: Letters are too small in the figures.

Figure 5: Give in full OTU.

L188: E. vexans.

Figure 6: Letters are too small in the figures.

Figure 7: Text in the figure can not be seen.

Figure 9: Colour explanation texts are too small.

L278: Cladosporium fulvum

L290: Colletotrichum - in italic

L296: anthracnose

L312: Materials

L324: Camellia sinensis var. sinensis - italic

References:

Journal abbreviations are not consistent.

Reviewer 3 Report

My comments can be found in the attached manuscript.

I understand that English is not your first language, and getting help from a native English speaker will improve the manuscript. There are a few typos in the manuscript. 

Round 2

Reviewer 3 Report

My comments can be seen in the attached MS.

The manuscript still needs some grammatical corrections; use 'present tense' in the MS.
